# FAST MNAS: UNCERTAINTY-AWARE NEURAL ARCHITECTURE SEARCH WITH LIFELONG LEARNING

## ABSTRACT

Sampling-based neural architecture search (NAS) always guarantees better convergence yet suffers from huge computational resources compared with gradient-based approaches, due to the *rollout bottleneck* – exhaustive training for each sampled generation on proxy tasks. This work provides a general pipeline to accelerate the convergence of the rollout process as well as the RL learning process in sampling-based NAS. It is motivated by the interesting observation that both the architecture and the parameter knowledge can be transferred between different experiments and even different tasks. We first introduce an uncertainty-aware critic (value function) in PPO to utilize the architecture knowledge in previous experiments, which stabilizes the training process and reduces the searching time by 4 times. Further, a life-long knowledge pool together with a block similarity function is proposed to utilize lifelong parameter knowledge and reduces the searching time by 2 times. It is the first to introduce block-level weight sharing in RL-based NAS. The block similarity function guarantees a 100% hitting ratio with strict fairness. Besides, we show a simply designed off-policy correction factor that enables "replay buffer" in RL optimization and further reduces half of the searching time. Experiments on the MNAS search space show the proposed FNAS accelerates standard RL-based NAS process by $\sim$10x (e.g. $\sim$256 2x2 TPUv2*days / 20,000 GPU*hour $\rightarrow$ 2,000 GPU*hour for MNAS), and guarantees better performance on various vision tasks.

## 1 INTRODUCTION

Neural architecture search (NAS) has made great progress in different tasks such as image classification (Tan & Le, 2019) and object detection (Tan et al., 2019b). And usually, there are four commonly used NAS algorithms: differentiable, one-shot, evolutional, and reinforcement learning (RL) based method. The RL-based method, due to its fair sampling and training processes, has often achieved a great performance among different tasks. However, one of the biggest challenges of it is the high demand for computing resources, which makes it hard to follow by the research community.

RL-based NAS consumes a large number of computing powers on two aspects: a) the need for sampling a large number of architectures to optimize the RL agent and b) the tedious training and testing process of these samples on proxy tasks. For example, the originator of NAS (Zoph & Le, 2016) requires 12,800 generations of architecture and current state-of-the-art MNAS (Tan et al., 2019a) and MobileNet-V3 (Howard et al., 2019) require 8000 or more generations to find the optimal architectures. Besides, each generation is usually trained for 5 epochs. All in all, it costs nearly 64 TPUv2 devices for 96 hours or 20,000 GPU hours on V100 for just one single searching process. With such a severe drawback, researchers start looking for other options like differential (Liu et al., 2018b; Chen et al., 2019), or one-shot based (Bender, 2019; Guo et al., 2019) method for NAS.

The one-shot family has drawn lots of attention recently due to its efficiency. It applies a single super-network based search space with that all the architectures, also called sub-networks, share parameters with the super-network during the training process. In this way, the training process is condensed from training thousands of sub-networks into training a super-network. However, this share-weight strategy may bring problems for the performance estimation of sub-networks. For example, two sub-networks may propagate conflicting gradients to their shared components, and the shared components may converge to favor one of the sub-networks and repel the other randomly. This

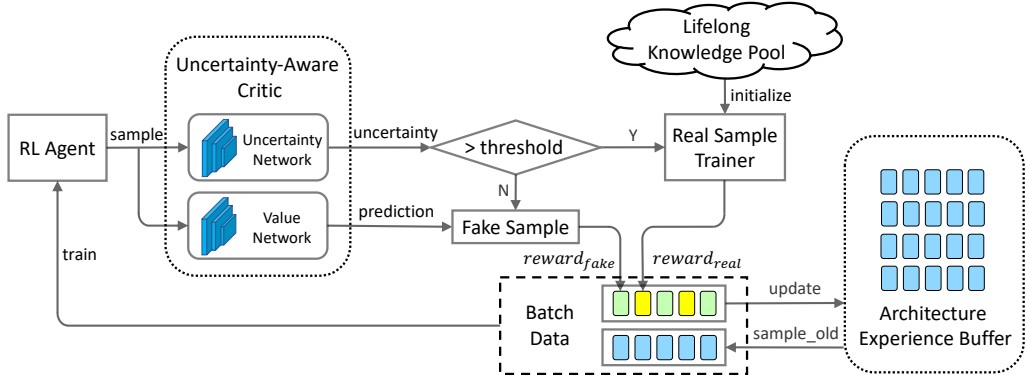

Figure 1: Reward along sample generation between FNAS and MNAS. Blue dots are the searching result of MNAS, while red dots are the results of FNAS.

Figure 2: The pipeline of FNAS. The proposed modules are highlighted in orange. Architectures are sampled by the RL agent and then passed to Uncertainty-Aware Critic (UAC) for predicted performance and the corresponding uncertainty. Then a decide module will determine whether the sample needs to be trained by Trainer. The Lifelong Knowledge Pool (LKP) helps to initialize new samples for training. Half of the samples in one batch come from Architecture Experience Buffer (AEB), the other half come from Trainer or UAC's Value Network.

conflicting phenomenon may result in instability of the search process and inferior final architectures, compared with RL-based methods.

In this work, we seek to combine the privilege of RL-based methods and one-shot methods, by leveraging the knowledge from previous NAS experiments. The proposed method is based on two **key observations**: First, the optimal architectures for different tasks have common architecture knowledge. Second, the parameter knowledge can also be transferred across experiments and even tasks.

Based on the observations, for transferable architecture knowledge, we develop **Uncertainty-Aware Critic (UAC)** to learn the architecture-performance joint distribution from other experiments even other tasks in an unbiased manner, utilizing the transferability of the *structural knowledge*, which reduces the sample's training time by 50% and the result is shown in Figure 1 (with UAC); For the transferable *parameter knowledge*, we propose **Lifelong Knowledge Pool (LKP)** to restore the block-level parameters and fairly share them to new samples' initialization, which speeds up each samples' convergence for 2 times, as shown in Figure 1 (with LKP); Finally, we also developed an **Architecture Experience Buffer (AEB)** with a significant off-policy correctness factor to store the old models for reusing in RL optimization, with half of the time saved. And this is shown in Figure 1 (with AEB). Under the strictly same environment as MNAS and MobileNet-v3, FNAS speed up the searching process by $10\times$ and the performances are even better.

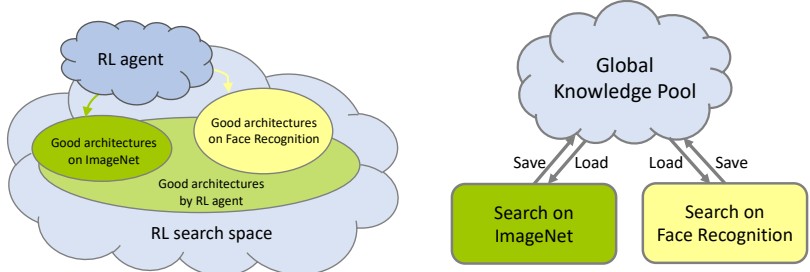

Figure 3: On the left, searching for neural architectures on different tasks leads to different optimal architectures. On the right, different tasks share the same global knowledge pool.

## 2 REVISITING SAMPLING-BASED NAS

### 2.1 NAS FAMILY

From the perspective of how to derive the performance estimation of an architecture, NAS methods can be split into two categories, sampling-based and share-weight based.

Sampling-based methods usually sample many architectures from the search space and train them independently. Based on the performance of these well-trained architectures, several ways can be utilized to fetch the best one, such as Bayesian optimization (Kandasamy et al., 2018), evolutionary algorithm (Real et al., 2019), and training an RL agent (Zoph & Le, 2016). The main drawback of these methods is a huge time and resource consumption of training the sampled architectures. To alleviate this issue, a common practice is to shorten the training epochs and use proxy networks with fewer filters and cells (Zoph et al., 2018; Tan et al., 2019a). Liu *et al.* (Liu et al., 2018a) propose to train a network to predict the final performance. Different from these methods, we leverage the accumulated knowledge to accelerate the training process.

Instead of training many architectures independently, the second kind of methods resorts to train a super-network and estimate the performance of architectures with weights shared from the super-network (Bender, 2019; Wu et al., 2019; Liu et al., 2018b; Chen et al., 2019; Xu et al., 2019; Cai et al., 2019; Stamoulis et al., 2019; Guo et al., 2019). With the easy access of performance estimation, DARTS (Liu et al., 2018b) proposes a gradient-based method to search for the best architecture in an end-to-end manner. However, as pointed in (Li & Talwalkar, 2019), the performance estimation based on shared weights may be unreliable. Chen *et al.* (Chen et al., 2019) proposes to progressively shrink the search space so that the estimation can be more and more accurate. Cai *et al.* (Cai et al., 2019) introduces a shrinking based method to train the supernet so as to generate networks of different scales without retraining. We also share weights between architectures but in different ways. We construct a general weight pool with many trained architectures, and when we want to train a new architecture, we initialize it by the trained architectures in the pool. In this way, the number of training epochs can be reduced without harming reliability.

## 3 KNOWLEDGE BETWEEN NAS EXPERIMENTS IS TRANSFERABLE

RL-based NAS consumes a lot of computing resources. MNAS, as it's said before, trains 8,000 models for the agent to converge, which costs 20,000 GPU hours on V100. And all the samples trained for one experiment will not be used anymore. However, the active differentiable-based NAS demonstrates that with various weight-sharing techniques, the NAS algorithms can be accelerated a lot. In this section, we will show that the knowledge of previous searched experiments can be reused by the following two observations, which helps the follow-up experiment greatly.

### 3.1 ARCHITECTURE KNOWLEDGE CAN BE TRANSFERRED

**Optimal architectures for different tasks have common architecture knowledge.** One always holds the assumption that the performance of a model is consistent among different tasks. A common

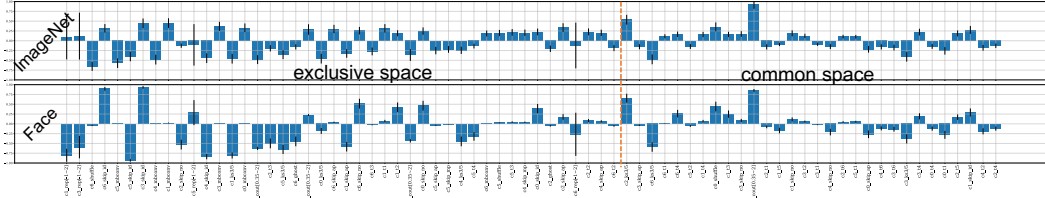

Figure 4: Expectation of each operator of optimal models of face experiment and ImageNet experiment. Calculated by the 100 optimal models of face experiments and ImageNet experiments and sorted by the significance of the difference.

practice of this assumption is applying good ImageNet (Deng et al., 2009) models to COCO object detection (Lin et al., 2014) as the backbone. In NAS, however, this assumption needs to be carefully checked as the huge search space of it requires the hypothesis to be well generalized. Here, We statistically verify this assumption.

In Figure 4, we sample 100 optimal models on face recognition and ImageNet classification tasks respectively. For each model, we firstly expand each digit of its embedding (i.e. a 35 dimension vector) to one-hot: e.g. from "4" to "1000"; "3" to "0100". In this way, the original embedding is expanded to 112-dim from 35-dim. After that, we calculate the expectation for each digit of this expended vector among the top 100 optimal architectures on the face recognition and ImageNet classification respectively. Shown in Figure 4, we get an observation that the operators can be divided into two spaces, i.e. **exclusive space**, where the probability difference is large and **common space**, where the probability difference is small.

Many previous works (Liu et al., 2018a; Kokiopoulou et al., 2019; Luo et al., 2018; Wen et al., 2019; Luo et al., 2020) use a predictor to predict a model's performance to speed up the NAS process. However, as the predictor requires thousands of samples to train, they usually implement in a progressive (Liu et al., 2018a) or semi-supervised manner (Luo et al., 2020). Inspired by the interesting observation above, we implement it in a unified way where different tasks' samples are used together to train a unified value network to predict a model's performance. When searching architecture on a new task, we just use directly the unified network trained by the old data and keep updating it in the new task during the search process, which speeds up the convergence of the value network. Showing in Figure 5a, when transferring a value network trained on ImageNet to face recognition task, the network converges much faster.

### 3.2 PARAMETER KNOWLEDGE CAN BE TRANSFERRED

Initializing the network by ImageNet pre-trained models and training the model on other tasks has nearly been a standard way as it can always speed up the convergence process. However, pretraining has been ignored in the NAS area as it may break the rank of different models. In our experiments, we observe that the trained checkpoint, we call it *parameter knowledge*, can help us to get the *real rank* faster than training from scratch. Besides, this feature holds regardless of the data distribution. We randomly sample 50 models and train them on ImageNet in two ways: from scratch or by initializing with *parameter knowledge* from face experiment. Then, we compare the rank correlation with real rank (i.e. fully trained rank) along the training process. Showing in Figure 5b and 5c, with *parameter knowledge* from face experiment, one gets more accurate rank in fewer epochs.

## 4 UNCERTAINTY-AWARE NAS WITH LIFELONG LEARNING

In this section, we introduce how we utilize the observations above to design three core modules to inherit common structural knowledge and parameter knowledge from task to task without bias.

### 4.1 UNCERTAINTY-AWARE CRITIC IN PPO

The value function is a common module and is widely used in RL algorithms like PPO but rarely used in traditional NAS. Usually, training a value function requires a large number of samples to

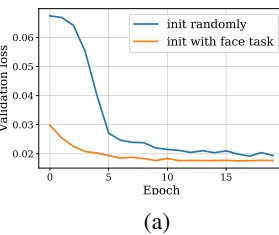 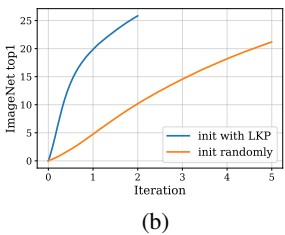 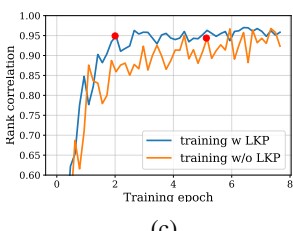

|(a)|(b)|(c)|

Figure 5: On the left, the value function pretrained on face recognition task converges much faster. On the right, Spearman rank-order correlation (Zar, 2005) along the training process of random initialization and block-level initialization.

converge (e.g. million-level steps in Atari env trained by ray[1]), which is unbearable for NAS as it means thousands of models needed to be trained and it's expensive. In our algorithm, alternatively, we propose the Uncertainty-Aware Critic to deal with this issue which is inspired by Section 3.

Given an architecture $A_i$ sampled from search space, a value network $V$ is utilized to predict the reward $V(A_i)$ of this sample, while $R(A_i)$ is the real reward of it. And the loss function to update $V$ is

$$L_V = |V(A) - R(A)| \tag{1}$$

Besides, an uncertainty network $U$ is utilized to predict the uncertainty $U(A_i)$ of this sample, which is used to learn discriminately whether a sample is in the distribution of learned samples. In our implementation, the loss function to update $U$ is

$$L_U = |U(A) - L_V| \tag{2}$$

If $U(A_i)$ is greater than a threshold, the sample may locate in the region which has not been learned by the value network. As a result, it will be trained from scratch to get its reward. Otherwise, one thinks the prediction $V(A_i)$ is accurate, and $V(A_i)$ will be regarded as the reward of $A_i$ to update the RL agent. The whole process is illustrated in Figure 2.

Samples need to be trained from scratch are defined as *real sample*, while others are defined as *fake sample* if its reward comes from $V$. With more *real samples* getting their rewards in our searched tasks, the value network becomes more accurate, thus the uncertainty value gets from $U$ will always decrease. Considering an extreme case where each sample in a batch gets a reward with low uncertainty and is classified as *fake samples*, the agent updated by these samples is easy to be over-fitted, which is not conducive to the exploration of the RL agent and it would lead to bad performance. In our implementation, we use the following constraints to balance the exploration and the exploitation of the RL agent to speed up its convergence without over-fitting.

- Constraint1: Sample whose uncertainty is higher than the threshold $\sigma$ needs to be trained from scratch.
- Constraint2: In each batch, when the number of *fake samples* is greater than 50% of the batch size, the extra *fake samples* will be thrown away and the agent will resample until enough *real samples* are gotten to fill the rest of the batch.

With these two constraints, we find that the algorithm can maintain a decent performance in an accelerated search process.

## 4.2 UNCERTAINTY-AWARE KERNEL POOL

*Parameter knowledge* can be transferred among different tasks to speed up the convergence of a network as shown in Section 3.2. However, traditional pretraining is not feasible in NAS, as there

---

[1]https://github.com/ray-project/rl-experiments

are thousands of different architectures in the search space and we can not afford to pretrain each architecture on a different task. To address this problem, we propose to initialize each architecture in a factorized way and use a fuzzy matching algorithm to guarantee the hit ratio. First, we define an architecture as a combination of n blocks $\{b_1, b_2, ..., b_n\}$. For any two architectures $A_i$ and $A_j$, although generally, their structures may be quite different, some of their blocks may be similar to each other, (eg. $b_2$ of $A_i == b_2$ of $A_j$), thus the weights of these parts could be shared. So we build a Kernel Pool to store all the previously trained models block by block in a key-value manner, where the key is the expand embedding of each block and the value is the *Parameter knowledge* of the block.

Recent research has found that fairness in weight sharing has a great impact on the final performance (Chu et al., 2019). So we apply the following two strategies to solve the problem of fairness.

- The checkpoints stored in the LKP are trained with equal iterations.
- For each block query, the proposed uncertainty function to ensure that the match ratio reaches more than 99%, which means less than 1% blocks have been unfairly initialized.

Given a query block $b_i$, we calculate the cosine similarity of the expanded embedding as in Section 3.1 between $b_i$ and each element in LKP. The block with the highest similarity will be recalled to initialize $b_i$. We show the overall process in Figure 3. Using LKP can speed up the search process by $2\times$. At the same time, in Section 3 we found that parameter knowledge is transferable. We apply the same LKP to different tasks and all the search processes are accelerated, illustrated in Figure 5b.

## 5 ARCHITECTURE EXPERIENCE BUFFER

In a general RL task, there are a lot of discussions about sample reuses. However, in RL-based NAS, sample efficiency is rarely mentioned. In our algorithm, we propose architecture experience buffer to store the sampled models in the form of architecture-performance pairs, and for each iteration in the future, the stored samples may be used again to update the RL agent to speed up its convergence. We call the samples stored in the experience buffer as *old samples* and the newly generated samples as *new samples*. Different from the traditional RL works, the proposed experience buffer has the following features:

- The buffer size is relatively small (usually 10 in our experiments). As the convergence of the RL agent is much faster than RL tasks, if the buffer size is set too large, the agent will focus on *old samples* and the convergence speed would be slow.
- In each batch, both *old samples* and *new samples* will be selected. To prevent the RL updating from biasing to the *old samples*, the percentage of the *old samples* in one batch is constrained to no more than 50%.

Some recent works (Schaul et al., 2015) suggested that the samples in the buffer should have different priorities. We do it similarly, defining different priorities in terms of their reward. Then, we sample from the buffer and reweight those samples with their priorities. For each sample $\{s_1, s_2, ..., s_n\}$ with their reward $R\{r_1, r_2..., r_n\}$ in AEB, the priority score is defined as: $P_i = \frac{exp(r_i)}{\sum_j exp(r_j)}$.

Following (Schaul et al., 2015), each sample will be reweighted by importance sampling weight. The reweighted score $S$ can be written in: $S_i = (N * P_i)^{-\beta}$, where $N$ is buffer size and $\beta$ is the annealing term and it will increase from 0 to 1 as the experiment proceeds.

## 6 FNAS ON VISION TASKS

In this section, we conduct different experiments on both ImageNet and million-level face recognition tasks to verify the effectiveness of FNAS. The details and results are as follows:

### 6.1 IMPLEMENTATION DETAILS

Following the standard searching algorithm as NASNet (Zoph et al., 2018), MNAS (Tan et al., 2019a) and AKD (Liu et al., 2019), we use an RNN-based agent optimized by PPO algorithm (Schulman

Table 1: Performance Results on ImageNet Classification. FNAS-Image×1.3 means scale up FNAS-Image for 1.3× along width.

| models | Type | FLOPs | Top1 Acc. (%) | Top5 Acc. (%) | Search Cost (GPU Hours) |
|---|---|---|---|---|---|
| MBv2 | Manual | 300M | 72 | 91 | 0 |
| ProxylessNAS | | 320M | 74.6 | 92.2 | 200 |
| DARTS | Share-weight | 574M | 73.3 | 91.3 | 96 |
| FairNAS | | 388M | 75.3 | 92.4 | 288 |
| Once-For-All | | 327M | 75.3 | 92.6 | 1200 |
| AmoebaNet | Evolutionary | 555M | 74.5 | 92 | 75,600 |
| MNAS | | 315M | 75.2 | 92.5 | 20,000 |
| NASNet | | 564M | 74 | 91.6 | 43,200 |
| MBv3 | RL-based | 219M | 75.2 | 91 | - |
| EfficientNetB0 | | 390M | 76.3 | 93.2 | - |
| **FNAS-Image** | | **225M** | **75.5** | **92.6** | **2000** |
| **FNAS-Image×1.3** | | **392M** | **77.2** | **93.5** | **2000** |

Table 2: Performance on MegaFace.

| model | FLOPs | Distractor num 1e5 | Distractor num 1e6 | Search Cost (GPU Hours) |
|---|---|---|---|---|
| MBv2 | 300M | 92.75 | 88.71 | 0 |
| ShuffleNet | 295M | 94.15 | 90.46 | 0 |
| MNAS | 313M | 93.41 | 89.47 | 20,000 |
| MBv3 | 218M | 94.15 | 90.64 | - |
| **FNAS-Face** | **227M** | **95.45** | **92.63** | **2000** |

Table 3: Performance on COCO.

| models | FLOPs | mAP |
|---|---|---|
| MNAS | 6.697 G | 27.68 |
| MBv2×1.0 | 6.675 G | 29.79 |
| MBv3×0.75 | 5.852 G | 29.25 |
| MBv3×1.0 | 9.060 G | 30.02 |
| **FNAS** | **8.021 G** | **30.44** |

et al., 2017). For ImageNet experiments, we sample 50K images from the training set to form the mini-val set and use the rest as the mini-training set. In each experiment, 8K models are sampled to update the RL agent. Note that when equipped with UAC or AEB, **not all samples need to be activated**, as many samples' rewards are directly returned from these two modules. For face experiments, we use MS1M (Guo et al., 2016) as the mini-training set, LFW (Huang et al., 2008) as the mini-val set. The final performance is evaluated on MegaFace (Kemelmacher-Shlizerman et al., 2016).

## 6.2 PROXYLESS FNAS ON IMAGENET

Just as MNAS has done, we also use a multi-objective reward to directly search on ImageNet. After the search process, we retrain the top 10 models with the largest reward near the target flops from scratch to verify the search results. In Table 1, we get a relatively higher result than the current SOTA network MBv3 (Howard et al., 2019). **Note that the model we search does not go through the pruning operation NetAdapt (Yang et al., 2018), which can reduce 10%∼15% computation and keep performance nearly unchanged.** Compared with EfficientNetB0, FNAS improves top 1 accuracy by 1 point under comparable computation budget. And still, there is nearly 10× of acceleration in the entire search process compared to MNAS (Tan et al., 2019a) or MBv3 (Howard et al., 2019).

## 6.3 PROXYLESS FNAS ON FINE-GRAINED FACIAL RECOGNITION

Besides verifying the performance of FNAS on ImageNet, we also test it on the fine-grained facial recognition task. As can be seen in Table 2, compared with MBv3, verification accuracy improves 2 points in comparable FLOPs under 1e6 distractors. When compared with MBv2, FNAS improves verification accuracy for nearly 4 points with 24% FLOPs reduction. The result shows: 1) FNAS has an obvious acceleration effect on different tasks and 2) the importance of searching directly on the target task.

Table 4: The effectiveness of the three proposed modules, MBv2×0.38 means scale up MBv2 for 0.38× along width

| Models | LKP | UAC | AEB | MFLOPs | Top1 Acc. (%) | Activated Samples | Search Epoches | Search Cost (GPU Hours) |
|---|---|---|---|---|---|---|---|---|
| MBv2×0.38 | | | | 81 | 62.65 | 0 | 0 | 0 |
| MNAS | | | | 72 | 64.23 | 8,000 | 1 | 4,000 |
| | | | | 74 | 65.19 | 10,000 | 4 | 20,000 |
| FNAS | ✓ | | | 75 | 64.97 | 8,000 | 1 | 4,000 |
| | ✓ | | | 76 | 65.22 | 8,000 | 2 | 10,000 |
| | | ✓ | | 72 | 64.28 | 2,300 | 1 | 1,150 |
| | | | ✓ | 72 | 64.44 | 4,500 | 1 | 2,250 |
| | ✓ | ✓ | ✓ | **85** | **66.25** | **2,000** | **1** | **1,000** |

Table 5: Transferability of UAC and LKP

| models | UAC or LKP | MFLOPs | Top1 Acc. (%) | Activated Samples | Search Epoches | Search Cost (GPU Hours) |
|---|---|---|---|---|---|---|
| MNAS | ✗ | 181 | 73.25 | 8,000 | 4 | 16,000 |
| **FNAS** | UAC init with face exp | **153** | **73.91** | **2,000** | **4** | **4,000** |
| MNAS | ✗ | 285 | 74.62 | 8,000 | 4 | 16,000 |
| **FNAS** | LKP init with face exp | **292** | **75.22** | **4,000** | **4** | **8,000** |

## 6.4 TRANSFERABILITY ON OBJECT DETECTION

We combine the model found on ImageNet in Table 1 with the latest pipeline of detection to verify its generalization. Table 3 shows the performance of the model on COCO (Lin et al., 2014). It can be seen that compared to MBv3, there is a significant improvement with our searched model.

## 7 ABLATION STUDY

### 7.1 THE EFFECTIVENESS OF THE THREE PROPOSED MODULES.

In this section, the effectiveness of UAC, LKP, AEB is verified when they are used alone or combined. Details are shown in Table 4. Three conclusions can be observed: 1. Sampling with LKP initialization gets real rank faster; 2. Fewer samples are required when NAS is equipped with UAC and AEB; and 3. 10× speedup can be achieved when NAS is equipped with LKP, UAC, and AEB.

### 7.2 THE TRANSFERABILITY OF THE PROPOSED MODULES.

In Section 3, we mentioned that knowledge between NAS experiments is transferable, which is also verified in the experiment. We use the UAC trained on the face as a pre-trained model and then transfer it to the ImageNet experiment. In the absence of 3/4 of activated samples, the optimal model surpasses baseline by 0.67% with fewer FLOPs, showing in Table 5. In addition, we use LKP with the checkpoints from face experiments and then search on ImageNet. In the absence of 1/2 activated samples, performance increases by 0.6%.

## 8 CONCLUSION

This paper proposes three modules (UAC, LKP, AEB) to speed up the entire running process of RL-based NAS, which consumes large amounts of computing power before. With these modules, fewer samples and less training computing resources are needed, making the overall search process 10× faster. We also show the effectiveness of applying those modules on different tasks such as ImageNet, face recognition, and object detection. More importantly, the transferability of UAC and LKP is being tested by our observation and experiments, which will guide us in tapping the knowledge of the NAS process.

ACKNOWLEDGMENTS

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

## A    APPENDIX

You may include other additional sections here.

