# OpenReview forum: "Fast MNAS: Uncertainty-aware Neural Architecture Search with Lifelong Learning"
_ICLR.cc/2021/Conference — Reject_

### Official Review · AnonReviewer3 · 2020-10-23
**A decent work**

**Rating:** 5
**Confidence:** 5

**Review:**

This paper proposes an RL-based neural architecture search approach to decrease the searching cost by introducing three modules to estimate uncertainty, restore parameters, and store old models. Compared to MNAS, it can significantly reduce the search cost up to x10, while giving competitive accuracy.

This paper is generally well-written and well-motivated, except for some unclear sentences;
-      Architecture knowledge is not well described. Compared to parameter knowledge, the authors should clarify what they are and the difference between them.
-      In Figure 4, it is unclear what the operators are and which operators are similar and different. Moreover, details are missing on how to sample 100 optimal models.
-      In Equation 1, the definition of the reward is missing.
-      LKP (the acronym first introduced in page 5) is not described.

Even if it accelerates the search process, it entails additional memory due to the proposed module; it stores learned networks. So, I think there’s a trade-off between search cost and the total memory we need to reserve. From this, I wonder reducing the search cost is more significant compared to increase the required memory.

In Table 3, why FNAS has higher FLOPs than MNAS? This should be properly elaborated.

In Table 4, the cases using two modules are missing. It would be great to see the results to see which component actually affects the performance.

---

> ### Author Response · Authors · 2020-11-24
> **Individual response**
>
> **Q1. This paper is generally well-written and well-motivated, except for some unclear sentences.**
>
> **A1.** Thank you for pointing out these problems, which may lead to an unclear understanding. We will improve these issues in subsequent versions.
>
> ​**Q2. Architecture knowledge is not well described.**
>
> **A2.** Each sampled architecture will be encoded into an embedding $E_R^d$ , dubbed 'architecture knowledge'. Each value in $E$ indicates an operator in the specific location of an architecture. At the searching process of NAS, we find that the top-k models ordered by reward on different tasks shown consistent operator selection in some indexes, showing as the common space in Figure 4.
>
> ​**Q3. ...it is unclear what the operators are and which operators are similar and different. Moreover, details are missing on how to sample 100 optimal models.**
>
> **Q3.** The pattern of the label of each bar is 'cx_y', where x means which block is, ranges from 0 to 6, and y means the operator's name. The detail of the operator will be added in the subsequent versions. 100 optimal models are top-k models ordered by reward.
>
> ​**Q4. In Equation 1, the definition of the reward is missing.**
>
> ​**A4.** For a fair comparison, we utilize the same reward function as MNAS[1].
> $$
>     R_a = ACC_a \times \frac{FLOPs_a}{T}
> $$
>  $ACC_a$ is the accuracy of architecture $a$, and FLOPs_a is the FLOPs of $a$ and $T$ is the target FLOPs.
>
>
> ​**Q5. LKP (the acronym first introduced in page 5) is not described.**
>
> **A5.** LKP is the acronym of Lifelong Knowledge Pool, which is first introduced in page 2. The title of section 4.2 'uncertainty-aware kernel pool' was a typo, which should be uncertainty-aware knowledge pool.
>
>
> **Q6. ...trade-off between search cost and the total memory. From this, I wonder reducing the search cost is more significant compared to increase the required memory.**
>
> **A6.** Yes, you are right, GPU memory for value network and uncertainty network(2 fully connected network with 200 hidden number) and the storage for trained checkpoints(29M per checkpoint) are minor cost compared to search cost.
>
> **Q7. why FNAS has higher FLOPs than MNAS?**
>
> **A7.** Thanks for pointing this out. We filled in the wrong data. The FLOPs of MNAS and  MBv2$\times$1.0  are 13.4G and 13.3G respectively.
>
> **Q8. The cases using two modules are missing...**
>
> **A8.** This is for simplicity as it is enough to prove the effectiveness of our proposed modules. We will release more comparative experiment results in a future version.
>
>
>
> [1] Tan, M., Chen, B., Pang, R., Vasudevan, V., Sandler, M., Howard, A. and Le, Q.V., 2019. Mnasnet: Platform-aware neural architecture search for mobile. In *Proceedings of the IEEE Conference on Computer Vision and Pattern Recognition* (pp. 2820-2828).

---

### Official Review · AnonReviewer4 · 2020-10-29
**interesting topic, lots of ideas, shallow integration of ideas, and unconvincing results**

**Rating:** 5
**Confidence:** 3

**Review:**

In this paper, the authors propose to use a sampling-based approach to neural architecture search, which combines a life-long knowledge pool, uncertainty aware critic, architecture experience buffer. This approach has been demonstrated with vision tasks involving days of TPU training.

Overall, I rank 5, marginally below the acceptance threshold. NAS is an underexplored topic. But the papers seems like an engineering project that combines multiple existing ideas from the others' work, and there lacks theoretical depth about clear mathematical formulation of the approach, and reasoning on why the approach should work.

Pros:

+ Having access to TPU
+ the NAS topic
+ Integration of lifelong learning, NAS, and several other ideas

Cons:

- Why lifelong learning should work, considering that we are not in multi-task learning and nonstationary environment scenario? How do we know that the experimental results is not some coincidence?
- Can we put down the whole framework mathematically? It seems that this paper has only two formulas about some loss functions.
- Can we reason about the math? For example, any ideas to better organize knowledge pool and ideas architecture experience buffer for a large number of architectures and parameters encountered?

---

> ### Author Response · Authors · 2020-11-24
> **Individual response**
>
> **Q1. Why lifelong learning should work. Can we put down the whole framework mathematically?**
>
> **A1.** The key contribution of our paper is the observation about the transferability of knowledge in NAS, which provides novel insight into the essence of NAS. Based on this observation, we utilize life long learning to demonstrate that this transferability can help to significantly shorten the search process of NAS. Moreover, we plan to explore the transferability of more tasks and provide a general knowledge pool to accelerate the search process on these tasks.
>
>
> **Q2. ...better organize knowledge pool and ideas architecture experience buffer for a large number of architectures and parameters encountered?**
>
> **A2.** The transferability in NAS is a new and under-explored research area, and our work just provides the key observation instead of a complete mathematical framework. But this is indeed a good direction deserving further exploitation. We are going to include more mathematical analysis in the next work.

---

### Official Review · AnonReviewer2 · 2020-10-30
**Official Blind Review**

**Rating:** 6
**Confidence:** 3

**Review:**

Summary: The paper propose a few improvements to the sampling-based NAS using RL: 1) an uncertainty-aware critic to decide whether the sample needs to be trained; 2) a life-long knowledge pool to initialize the sample that needs training; and 3) an architecture experience buffer to reuse old samples for RL training. The experiments are done on ImageNet, facial recognition and transferability on object detection. The proposed methods are compared with related works. Finally the paper finishes with ablation studies on both the effectiveness and transferability of the proposed modules.

Strengths:
- The paper is well-written with clear flow and structures.
- The three proposed modules are novel and improve the search cost significantly while achieving better performance.
- It's great to see the authors has done a comprehensive comparison with the related methods for multiple tasks. The ablation study also demonstrate the effectiveness of the three proposed modules.

Weakness:
- The improvement of Top1 Acc. on ImageNet is marginal (without the 1.3 scale up) and worse than some of the recently proposed differentiable NAS work which requires far less search cost (comparable to DARTS).
- There could be more discussion on related work studying uncertainty in RL or in supervised learning, given that it is one of the core modules in the proposed pipeline and uncertainty an important topic in general.

---

> ### Author Response · Authors · 2020-11-24
> **Individual response**
>
> **Q1. The improvement of Top1 Acc. on ImageNet is marginal**
>
> **A1.** Our core contribution is not a new search space or architecture, but a general RL-based search pipeline, which speeds up the traditional RL-based method 10$\times$. Our search setting follows the implementation of MNAS[1] and MBV3[2]. With 10 times less search cost, we still get better results compared with MNAS and MBV3. For example, FNAS-Image is 0.3\% more accurate on ImageNet classification while reducing computation by 28\% compared to MNAS. FNAS-Image$\times$1.3 is 0.9\% more accurate than EfficientNetB0[3] on ImageNet classification in comparable computation cost.
>
> **Q2. There could be more discussion on related work studying uncertainty in RL or in supervised learning.**
>
> **A2.** Uncertainty is commonly used to balance the exploration and exploitation in RL tasks [4] [5] [6]. RL agent has more exploration for samples with high uncertainty. In the paper, the uncertainty network is used to judge the inaccuracy of the prediction of the value network. The predicted uncertainty is used to determine whether to train an architecture from scratch, instead of changing the sampling strategy.
>
> In supervised learning, uncertainty has different meanings in various tasks. For example in face recognition, uncertainty is utilized to indicate whether a training sample is a noise or not [7]. While in object detection, uncertainty is used to represent the confidence about a predicted bounding box[8]. Uncertainty is also employed in the generative adversarial network to substitute the full supervision with the attribute ratings learned from weak supervisions[9]. The uncertainty in this paper represents different meanings and is used in different ways compared with these examples of supervised learning.
>
> [1] Tan, M., Chen, B., Pang, R., Vasudevan, V., Sandler, M., Howard, A. and Le, Q.V., 2019. Mnasnet: Platform-aware neural architecture search for mobile. In *Proceedings of the IEEE Conference on Computer Vision and Pattern Recognition* (pp. 2820-2828).
>
> [2] Howard, A., Sandler, M., Chu, G., Chen, L.C., Chen, B., Tan, M., Wang, W., Zhu, Y., Pang, R., Vasudevan, V. and Le, Q.V., 2019. Searching for mobilenetv3. In *Proceedings of the IEEE International Conference on Computer Vision* (pp. 1314-1324).
>
> [3] Tan, M. and Le, Q.V., 2019. Efficientnet: Rethinking model scaling for convolutional neural networks. *arXiv preprint arXiv:1905.11946*.
>
> [4] Chapelle, O. and Li, L., 2011. An empirical evaluation of thompson sampling. In *Advances in neural information processing systems* (pp. 2249-2257).
>
> [5] Hao, B., Abbasi Yadkori, Y., Wen, Z. and Cheng, G., 2019. Bootstrapping upper confidence bound. *Advances in Neural Information Processing Systems*, *32*, pp.12123-12133.
>
> [6] Auer, P., 2002. Using confidence bounds for exploitation-exploration trade-offs. *Journal of Machine Learning Research*, *3*(Nov), pp.397-422.
>
> [7] Chang, J., Lan, Z., Cheng, C. and Wei, Y., 2020. Data Uncertainty Learning in Face Recognition. In Proceedings of the IEEE/CVF Conference on Computer Vision and Pattern Recognition (pp. 5710-5719).
>
> [8] He, Y., Zhu, C., Wang, J., Savvides, M. and Zhang, X., 2019. Bounding box regression with uncertainty for accurate object detection. In Proceedings of the IEEE Conference on Computer Vision and Pattern Recognition (pp. 2888-2897).
>
> [9] Han, L., Gao, R., Kim, M., Tao, X., Liu, B. and Metaxas, D.N., 2020. Robust Conditional GAN from Uncertainty-Aware Pairwise Comparisons. In AAAI (pp. 10909-10916).

---

### Official Review · AnonReviewer1 · 2020-11-01
**Practical approach based on real observation but less technical depth**

**Rating:** 6
**Confidence:** 4

**Review:**

Summary
-------

This paper proposes a fast general framework (FNAS) for neural architecture search (NAS) problem to enhance the processing efficiency up to 10x times. Three interesting strategies (UAC, LKP, AEB) for reinforcement learning (RL) processing are introduced in the proposed FNAS and evaluated by extensive experiments to show their efficacy. In particular, the assumption that architecture knowledge is transferable has been verified by real observation.

However, the authors paid more attention to introduce the fact based on observations and the thoughts of the framework design, thus neglected the technical depth for the key component (UAC) that has highest impact on the overall performance.


Strengths
---------

- The paper is well-organized and well-written thus easy to understand, including motivation, approach, and experiments.

- The proposed framework (FNAS) is general, practical and convincing. The three strategies (UAC, LKP, AEB) for RL processing are based on real observations shown in Figure 4, which positively supports the motivation of the approach.

- The evaluations are conducted on extensive experiments with solid results including ablation studies for the three different components LKP, UAC, and AEB (although it might be not sufficient; see next the weaknesses).


Weaknesses
----------

- The technical depth was neglected. For example, the  Uncertainty-Aware Critic (UAC) should be considered as the key component of FNAS framework because it was shown that the UAC has highest impact on (biggest contribution to) the overall performance in terms of efficiency in Table 4. However, there is no any technical/mathematical introductions about how the uncertainty network $U$ is obtained/prepared, and how the $U$ contribute to the NAS in detail.

- Discussions regarding possible over-fitting are not sufficient. For example, the constraint (threshold $\delta$) of uncertainty is introduced into the UAC strategy (Sec. 4.1), but without any experiment results to show the impact of such hyperparameters in the FNAS framework that has to be considered as trade-off parameter against over-fitting effects.

- Similarly, in the AEB strategy (Sec. 5), the buffer size ($N$), and the annealing term ($\beta$) are hyperparameters that should have impact on the over-fitting effects during RL processing. However, the authors did not provide any testing results to confirm their impacts. For example, why did the authors determine the buffer size $N=10$ in the experiments?

- The testing of FNAS on vision tasks are not sufficient. This paper provided results on classification (ImageNet) and face recognition tasks, but how about other tasks such as object detection, tracking, person re-identification, and segmentation?


Other Questionable Points
-------------------------

- The loss functions shown in Eq. (1) and Eq. (2) seem too simple. Are they really sufficient to get high performance? Is there any potential loss functions or improvements that would get better performance?

- In Table 1, why the numbers of "GPU Hours" for MBv3 and EfficientNetB0 are not shown? It is inconsistent with textual description in Sec. 6.2 ("there is nearly 10x of acceleration ...") that is not able to confirm.

- In Table 1 and 2, regarding the numbers of "GPU Hours" like 20,000 and 2,000, do they indicate the real runtime in the experiments or only estimated values? As we know, 20,000 hours are roughly equal to 2.3 years, and 2,000 hours are similarly equal to 2.8 months. The reliability of the experiment results might be doubting.

- In the references, there are too many informal publications cited from arXiv. Instead, they should be replaced by their formal publications at the corresponding conferences or journals.

---

> ### Author Response · Authors · 2020-11-24
> **Individual response**
>
> **Q1. The technical depth was neglected... there is no any technical/mathematical introductions about how the uncertainty network is obtained/prepared, and how the U contribute to the NAS in detail.**
>
> **A1.** The core contribution UAC comes from our observation that *architecture knowledge can be transferred*. To utilize the *architecture knowledge*, we use $V$ to learn the distribution of samples' performance. Besides, an uncertainty network learns discriminately whether a sample is in the distribution of learned samples. For simplicity, We define uncertainty as the L1 distance between the prediction of the value network and the real reward. A simple 2 layers MLP is utilized to predict the uncertainty. In the search process, we will use the uncertainty network to determine whether the architecture needs to be trained from scratch. Besides, all newly trained samples will be used to update the uncertainty network. The missed details will be added to the next version.
>
> **Q2. Discussions regarding possible over-fitting are not sufficient.**
>
> **A2.** Good question.
> In Section4.1 and Section 5, we mentioned the setting of hyperparameters. If the threshold $\sigma$ is set to 0, all samples need to be trained from scratch without any acceleration effect. If the threshold is set to 1, the RL-agent will over-fit to the fake samples. Therefore, we set an intermediate value based on the search log to ensure 2 times speedup while avoiding the risk of over-fitting. To guarantee the convergence speed of the RL agent, we set the buffer size $N$ to a relatively small value and ensure reuse times of each sample is no more than 2. $\beta$ is the annealing term and it will increase from 0 to 1 as the experiment proceeds, following the original implementation of prioritized experience replay[1]. We set those hyperparameters empirically, and did not fine-tune these parameters based on searched tasks, as adjusting the parameters during the NAS process requires a lot of computing resources. But as you said, these parameters may have a certain influence on over-fitting. We will add the impact of these parameters on the final convergence and put it in the next version.
>
>
>
> **Q3. The testing of FNAS on vision tasks are not sufficient.**
>
> **A3.** For fairly comparison, we follow the general practice of MNAS[2] and MBV3[3], which implements searching experiments on ImageNet task, and transfer the searched architecture to other general vision tasks, objection detection in our paper. Experiments in other vision tasks will be added and released soon. Besides, to verify the generalization of our proposed modules, we also conduct a search experiment on face recognition tasks, which uses the largest dataset.
>
>
>
>
> ### Other Questionable Points
>
> **Q1. loss functions ... seem too simple**
>
> **A1.** Our core contribution is the whole acceleration pipeline, instead of a new loss function to train UAC. We directly use the simple L1 loss to train the value network and uncertainty network, as those can already get decent performance. Other loss function may lead to higher performance, such as smooth L1 loss [4], which is worthy of follow-up exploration.
>
> **Q2. why the numbers of "GPU Hours" for MBv3 and EfficientNetB0 are not shown. It is inconsistent...**
>
> **A2.** In these two papers, the search cost is not explicitly mentioned. Because MBV3 uses the same search algorithm as MNAS, we assume that they consume approximate computing resources to search.
>
> **Q3. In Table 1 and 2, regarding the numbers of "GPU Hours" like 20,000 and 2,000, do they indicate the real runtime in the experiments or only estimated values?**
>
> **A3** This GPU hour is the time converted to a single GPU. But the entire search process can be parallelized, so we will use many GPUs to search at the same time. Suppose I use 100 GPUs to search at the same time, the entire search time will take 200 hours for MNAS. For FNAS, it only takes 20 hours.
>
> **Q4. In the references, there are too many informal publications cited from arXiv.**
>
> **A4.** we will improve these issues in subsequent versions.
>
>
> [1] Horgan, D., Quan, J., Budden, D., Barth-Maron, G., Hessel, M., Van Hasselt, H. and Silver, D., 2018. Distributed prioritized experience replay. In *International Conference on Learning Representations*.
>
> [2] Tan, M., Chen, B., Pang, R., Vasudevan, V., Sandler, M., Howard, A. and Le, Q.V., 2019. Mnasnet: Platform-aware neural architecture search for mobile. In *Proceedings of the IEEE Conference on Computer Vision and Pattern Recognition* (pp. 2820-2828).
>
> [3] Howard, A., Sandler, M., Chu, G., Chen, L.C., Chen, B., Tan, M., Wang, W., Zhu, Y., Pang, R., Vasudevan, V. and Le, Q.V., 2019. Searching for mobilenetv3. In *Proceedings of the IEEE International Conference on Computer Vision* (pp. 1314-1324).
>
> [4] Girshick, R., 2015. Fast r-cnn. In *Proceedings of the IEEE international conference on computer vision* (pp. 1440-1448).

---

### Decision · Program_Chairs · 2021-01-07
**Final Decision**

**Decision:**

Reject

**Comment:**

This paper presents a compelling mechanism for reducing the neural architecture search process based on accumulated experience  that the reviewers found compelling with significant improvements in performance.  This is an intriguing idea. However, there were concerns about clarity that need to be addressed, and more concerning, the paper lacked technical depth or details in several aspects described in the reviews.  The authors subsequent response and revisions have somewhat addressed these issues.

The reviewer discussion had mixed opinions, with some for weak acceptance and others for weak rejection.  There were compelling points that the contribution is significant, but overall this paper would benefit from thoroughly addressing the shortcomings mentioned in the reviews before it is ready for publication.